# TEACHER GUIDED ARCHITECTURE SEARCH

## ABSTRACT

Strong improvements in neural network performance in vision tasks have resulted from the search of alternative network architectures. Prior work has shown that this search process can be automated and guided by evaluating candidate network performance following limited training (Performance Guided Architecture Search or PGAS). However, because of the large architecture search spaces and the high computational cost associated with evaluating each candidate network, further gains in computational efficiency are needed. Here we present a method termed *Teacher Guided Search for Architectures by Generation and Evaluation (TG-SAGE)* that produces up to an order of magnitude in search efficiency over PGAS methods. Specifically, TG-SAGE guides each step of the architecture search by evaluating the similarity of internal representations of the candidate networks with those of the (fixed) *teacher* network. We show that this procedure leads to significant reduction in required per-sample training and that, this advantage holds for two different search spaces of architectures, and two different search algorithms. We further show that in the space of convolutional cells for visual categorization, TG-SAGE finds a cell structure with similar performance as was previously found using other methods but at a total computational cost that is *two orders of magnitude* lower than Neural Architecture Search (NAS) and more than *four times* lower than progressive neural architecture search (PNAS). These results suggest that TG-SAGE can be used to accelerate network architecture search in cases where one has access to some or all of the internal representations of a *teacher* network of interest, such as the brain.

## 1 INTRODUCTION

The accuracy of deep convolutional neural networks (CNNs) for visual categorization has advanced substantially from 2012 levels (AlexNet (Krizhevsky et al., 2012)) to current state-of-the-art CNNs like ResNet (He et al., 2015), Inception (Szegedy et al., 2014), DenseNet (Huang et al., 2016). This progress is mostly due to discovery of new network architectures. Yet, even the space of feedforward neural network architectures is essentially infinite and given this complexity, the design of better architectures remains a challenging and time consuming task.

Several approaches have been proposed to automate the discovery of neural network architectures, including random search (Pinto et al., 2009), reinforcement learning (Zoph & Le, 2017), evolution (Real et al., 2016), and sequential model based optimization (SMBO) (Liu et al., 2017; Bergstra et al., 2012a). These methods operate by iteratively sampling from the hyperparameter space, training the corresponding architecture, evaluating it on a validation set, and using the search history of those scores to guide further architecture sampling. But even with recent improvements in search efficiency, the total cost of architecture search is still outside the reach of many groups and thus impedes the research in this area (e.g. some of the recent work in this area has spent 40-557k GPU-hours for each search experiment (Real et al., 2018; Zoph & Le, 2017)).

What drives the total computational cost of running a search? For current architectural search procedures (above), the parameters of each sampled architecture must be trained before its performance can be evaluated and the amount of such training turns out to be a key driver in the total computational cost. Thus, to reduce that total cost, each architecture is typically only partially trained to a *premature* state and its *premature* performance is used as a proxy of its *mature* performance (i.e. the performance it would have achieved if was actually fully trained).

Because the search goal is high *mature* performance in a task of interest, the most natural choice of an architecture evaluation score is its *premature* performance. However, this may not be the best choice of evaluation score. For example, it has been observed that, as a network is trained, multiple sets of internal features begin to emerge over network layers, and the quality of these internal features determines the ultimate behavioral performance of the neural network as a whole. Based on these observations, we reasoned that, if we could evaluate the quality of a network's internal features even in a very *premature* state, we might be able to more quickly determine if a given architecture is likely to obtain high levels of *mature* performance.

But without a reference set of high quality internal features, how can we determine the quality of a network's internal features? The main idea proposed here is to use features of a high performing "teacher" network as a reference to identify promising sample architectures at a much earlier *premature* state. Our proposed method is inspired by prior work showing that the internal representations of a high-performing *teacher* network can be used to optimize the parameters of smaller, shallower, or thinner student networks (Ba & Caruana, 2014; Hinton et al., 2015; Romero et al., 2014). It is also inspired by the fact that such internal representation measures can potentially be obtained from primate brain and thus could be used as an ultimate teacher. While our ability to simultaneously record from large populations of neurons is fast growing (Stevenson & Kording, 2011), these measurements have already been shown to have remarkable similarities to internal features of CNNs (Yamins et al., 2014; Schrimpf et al., 2018).

We refer to this method as Teacher Guided Search for Architectures by Generation and Evaluation (TG-SAGE). Specifically, TG-SAGE guides each step of an architecture search by evaluating the similarity of several internal feature representations of each sampled architecture with those of a fixed, high-performing teacher network with unknown architectural parameters but observable internal states. We found that when this evaluation is combined with the usual performance evaluation (above), we can predict the mature performance of sampled architectures with an order of magnitude less *premature* training and thus an order of magnitude less total computational cost. We then use this observation to execute multiple runs of TG-SAGE for different architecture search spaces to confirm that TG-SAGE can indeed discover network architectures of comparable mature performance to those discovered with performance-only search methods, but with far less total computational cost.

## 2 PREVIOUS WORK

There have been several studies on automatic design of neural network architectures in the past few years. Real et al. (Real et al., 2016; 2018) used an evolutionary approach in which samples taken from a pool of networks were engaged in a pairwise competition game. This method searched for optimal architectures and weights jointly by reusing all or part of weights from the parent network in an effort to reduce the computation cost associated with training the candidate networks as well as the final retraining of the best found networks. However, it is not clear to what degree this procedure has cut down on the computational cost compared to alternative search method that depend on (some) training for each candidate network starting from an initial point. There have also been several studies on using reinforcement learning in agents that learn to design high performing neural network architectures (Baker et al., 2016; Zoph & Le, 2017). Of special relevance to this work is Neural Architecture Search (NAS) (Zoph & Le, 2017) in which a long short-term memory network (LSTM) trained using REINFORCE was used to learn to design neural network architectures for object recognition and natural language processing tasks. A variation of this approach was later used to design convolutional cell structures similar to those used in Inception network that could be transferred to larger datasets like Imagenet (Zoph et al., 2017).

While most of these works have focused on discovering higher performing architectures, there has been a number of efforts emphasizing the computational efficiency in hyperparameter search. In order to reduce the computational cost of architecture search, Brock et al. (Brock et al., 2017) proposed to use a hypernetwork (Ha et al., 2016) to predict the layer weights for any arbitrary candidate architecture instead of retraining from random initial values. Hyperband Li et al. (2017) formulated hyperparameter search as a resource allocation problem and improved the efficiency by controlling the amount of resources (e.g. training) allocated to each sample. Similarly, several other methods proposed to increase the search efficiency by introducing early-stopping criteria during

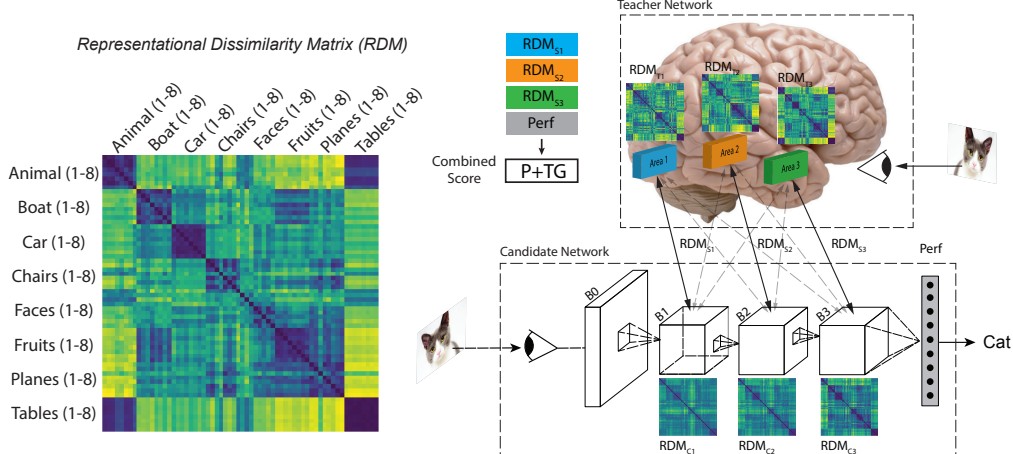

Figure 1: Overview of TG-SAGE method. left – Demonstration of an exemplar RDM matrix for a dataset with 8 object categories and 8 object instances per category. right – overview of TG-SAGE method. Correlation between RDMs of candidate and teacher networks, are combined with candidate network premature performance to form P+TG score for guiding the architecture search.

training Baker et al. (2017) or by extrapolating the learning curve Domhan et al. (2015). These approaches are closely related to our proposed method in that, their main focus is to reduce the per-sample training cost.

Some recent work attempted to jointly optimize for the network hyperparameters as well as the trainable weights themselves. While this is a very interesting idea that significantly reduces the computational cost of architecture search, in its current form it can only be applied to the spaces of network architectures in which the number of trainable weights do not change as a result of hyperparameter choices (e.g. when the number of filters in a CNN is fixed). Efficient NAS (Pham et al., 2018) and DARTS (Liu et al., 2018) methods proposed to share the trainable parameters across all candidate networks and to jointly optimize for the hyperparameters and the network weights during the search. While these approaches led to significant reduction in total search cost, they did so by constraining the search space due to considerations regarding the shared trainable weights. More recently progressive neural architecture search (PNAS) (Liu et al., 2017) proposed a sequential model based optimization (SMBO) approach that learned a predictive model of performance given the hyperparameters through a procedure which gradually increased the complexity of the space. This approach led to an impressive $20\times$ improvement in the computational cost of search compared to NAS.

## 3 METHODS

### 3.1 REPRESENTATIONAL DISSIMILARITY MATRIX:

Representational Dissimilarity Matrix (RDM) Kriegeskorte et al. (2008) is a statistic computed for a representation space that quantifies the dissimilarity between activity patterns in that space in response to pairs of inputs or input categories. For a given feature matrix $F \in \mathbb{R}^{n_i \times n_f}$ which contains $n_f$ features measured in response to $n_i$ images, we derive RDM ($M^F$) by computing the pairwise distances between each pair of feature vectors (rows in $F$) using a distance measure like correlation residual.

$$M^F \in \mathbb{R}^{n_c \times n_c}, M_{i,j} = 1 - corr(F_i^c, F_j^c) \qquad (1)$$

When calculating RDM for object categories (instead of individual images) we substitute the matrix $F$ with $F^c$ in which each row $c$ contains the average activity pattern across all images of category $c$. Once RDM is calculated for two representation spaces, we can evaluate the similarity of those spaces by calculating the correlation coefficient (e.g. Pearson's $r$) between the two RDM matrices.

## 3.2 Representational Similarity with a Teacher Network as Surrogate

The largest portion of cost associated with neural network architecture search comes from training the sampled networks, which is proportional to the number of training steps (SGD updates) performed on the network. Due to the high cost of fully training each sampled network, in most cases a surrogate score is used as a proxy for the *mature* performance. Correlation between the surrogate and *mature* score may affect the architecture search performance as poor proxy values could guide the search algorithm towards suboptimal regions of the space. Previous work on architecture search in the space of Convolutional Neural Networks (CNN) have concurred with the empirical surrogate measure of *premature* performance after about 20 epochs of training. While 20 epochs is much lower than the usual number of epochs used to fully train a CNN network (300-900 epochs), it still forces a large cost on conducing architecture searches. We propose that evaluating the internal representations of a network would be a more reliable measure of architecture quality during the early phase of training (e.g. after several hundreds of SGD iterations), when the features are starting to be formed but the network is not yet performing reliably on the task.

An overview of the procedure is illustrated in Figure-1. We evaluate each sampled model by measuring the similarity between its RDMs at different layers (e.g. $RDM_{C1-C3}$) to those extracted from the teacher network (e.g. $RDM_{T1-T3}$). To this end, we compute RDM for all or a subset of layers in the network and then compute the correlation between all pairs of student and teacher RDMs. To score a candidate network against a given layer of the teacher network, we consider the highest RDM similarity to teacher layer calculated over all layers of the student network (e.g. $RDM_{S1-S3}$). Finally, we construct an overall score by taking the mean of the RDM scores which we call TG (Teacher Guidance). We also define a combined Performance + TG (P+TG) score which is formulated as weighted sum of *premature* performance and TG score in the form of $P + \alpha TG$. In this fashion, the combined validation score guides the architecture search to maximize performance as well as representational similarity with the teacher architecture. We consider the teacher architecture as any high-performing network with unknown architecture but observable activations. We can have one or several measured endpoints from the teacher network that each could potentially be used to generate a similarity score.

## 4 Experiments and Results

### 4.1 Teacher Guidance for Architecture Search

We first investigated if a teacher similarity evaluation measure (P+TG) of premature networks improves the prediction of *mature* performance (compared to evaluation of only *premature* performance, P). To do this, we made a pool of CNN architectures for which we computed the *premature* and *mature* performances as well as the *premature* RDMs (a measure of the internal feature representation, see 3.2) at every model layer. We populated the pool by selecting the top 5 network architectures found at a range of checkpoints (every 100 samples) of the performance guided architecture searches. We included architectures (n=116) from two RL searches for 10-layer and another two for 20-layer CNNs with 20 epoch/sample training (see description of search spaces in the supp. material). In this way we included sample networks with a wide range of performance that also included the best network architectures found during each search.

In experiments throughout this paper, we used a variant of ResNet (He et al., 2015) with 54 convolutional layers ($n = 9$) as the teacher network. This architecture was selected as the *teacher* because it is high performing (top-1 accuracy of 94.75% and 75.89% on CIFAR10 and CIFAR100 datasets respectively). Notably, the teacher architecture is not in our search spaces (see supp. material). The features (after each of the three stacks of residual blocks, here named L1-L3) were chosen as the teacher's internal features, and a RDM was created from each using random subsample of features in that layer. We did not attempt to optimize this choice  these were chosen simply because they sampled approximately evenly over the full depth of the teacher.

We found that the earlier teacher layers (L1) are better predictors of the *mature* performance compared to other layers early on during the training (<2epochs) but as the training progresses, the later layers (L2 and L3) become better predictors (~3epochs) and with more training (>3epochs) the *premature* performance becomes the best single predictor of the *mature* (i.e. fully trained) performance. However the combined "P+TG" score (see 3.2) composes the best predictor of *mature* performance

during most of the training period (Figure 2-right). This observation was consistent with previous findings that learning in deep networks predominantly happen "bottom-up" (Raghu et al., 2017).

In order to find the optimum TG weight factor, we varied the $\alpha$ parameter (section 3.2) and measured the change in correlation between the P+TG score and the *mature* performance (see Figure 3 in supplementary material). We found that for networks trained for ~2 epochs, a value of $\alpha = 1$ is close to optimum. In addition to ResNet, we also analyzed a second teacher network, namely NASNet (see section 2 in supp. material) and confirmed our findings using the alternative teacher network. We also found that NASNet features (which performs higher than ResNet; 82.12% compared to 75.9%) form a better predictor of *mature* performance in almost all training regimes (Figure 4 in supp. material).

## 4.2 Teacher Guided Search in the Space of Convolutional Networks

As outlined in the Introduction, we expected that the (P+TG) evaluation score's improved predictivity (Figure 2) should enable it to support a more efficient architecture search than performance evaluation alone (P). To test this directly, we used the (P+TG) evaluation score in full architectural search experiments using a range of configurations. For these experiments, we searched two spaces of convolutional neural networks similar to previous search experiments (Zoph & Le, 2017) (maximum network depth of either 10 or 20 layers). These architectural search spaces are important and interesting because they are large. In addition, because networks in these search spaces are relatively inexpensive to train to maturity, we could evaluate the true underlying search progress at a range of checkpoints (below). We ran searches in each space using four different search methods: using the (P+TG) evaluation score at 2 or 20 epochs of premature training, and using the (P) evaluation score at either 2 or 20 epochs of premature training. For these experiments, we used Random, RL, as well as TPE architecture selection algorithm (see Methods), and we halted the search after 1000 or 2000 sampled architectures (for the 10- and 20-layer search spaces, respectively). We conducted our search experiments on CIFAR100 instead of CIFAR10 because of larger number of classes in the dataset that provided a higher dimensional RDM.

Table 1 summarizes results of our search experiments. We found that, for all search configurations, the (P+TG) driven search algorithm (i.e. TG-SAGE) consistently outperformed the performance-only driven algorithm (P) in that, using equal computational cost it always discovered higher performing networks. This gain was substantial in that TG-SAGE found network architectures with approximately the same performance as (P) search but at $\sim 10\times$ less computational cost (2 vs. 20 epochs; Table 1).

To assess and track the efficiency of these searches, we measured the maximum validation set performance of the fully trained network architectures returned by each search as its current choice of the top-5 architectures. We repeated each search experiment three times to estimate variance in these measures resulting from both search sampling and network initial filter weight sampling. Figure 3

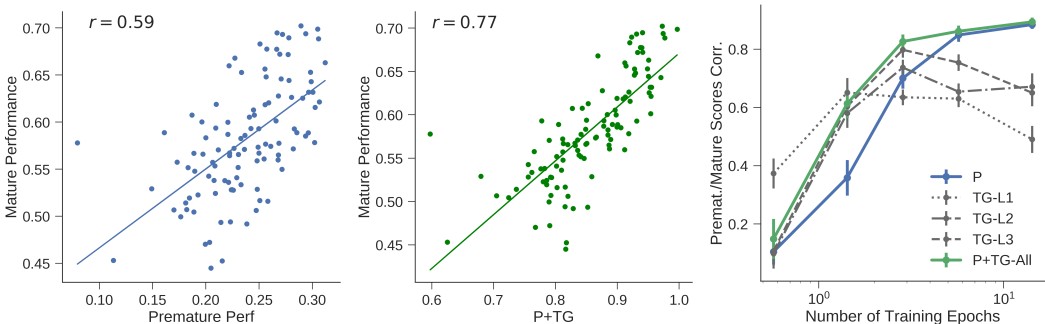

Figure 2: Comparison of performance and P+TG measures at premature state (epochs=2) as predictors of mature performance. (left) scatter plot of premature and mature performance values. (middle) scatter plot of premature P+TG measure and mature performance. (right) Correlation between performance, single layer RDMs, and combined P+TG measures with mature performance at varying number of premature training epochs.

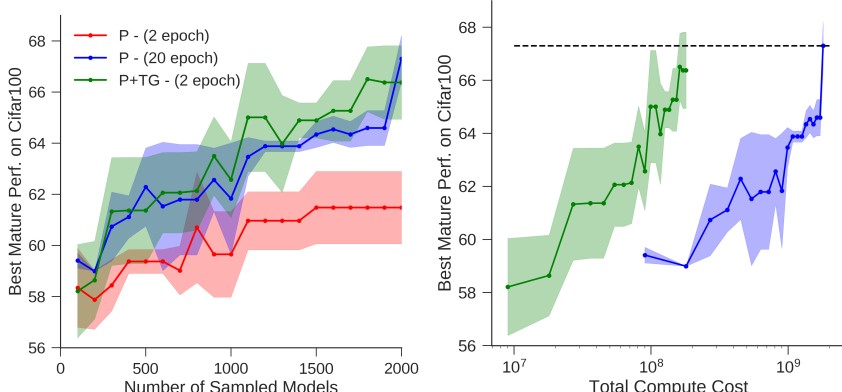

Figure 3: Effect of different surrogate measures on architecture search performance. (left) shows the average C100 performance of the best network architectures found during different stages of three runs of RL search in each case (see text). (right) same as the plot on left but displayed with respect to the total computational cost invested (number of training images × number of epochs × number of samples).

shows that the teacher guided search (P+TG) leads to finding network architectures that were on par with performance guided search (P) throughout the search runs while being 10× more efficient.

### 4.3 TEACHER GUIDED SEARCH IN THE SPACE OF CONVOLUTIONAL CELLS

In order to find architectures that are transferable across datasets we applied TPE search algorithm with P+TG score to the space of convolutional cells similar to the one used in (Liu et al., 2017). After a cell structure is sampled, the full architecture is constructed by stacking the same cell multiple times with a predefined structure (see supplementary material). While both RL and TPE search methods led to similar outcomes in our experiments in section 4.1, average TPE results were slightly higher for both experiments. Hence, we chose to conduct the search experiment in this section using TPE algorithm with the same setup as in section 4.1 using CIFAR100 with 1000 samples.

For each sample architecture, we computed RDMs for each cell's output. Considering that we had $N = 2$ cell repetitions in each block during search, we ended up with 8 RDMs in each sampled cell that were compared with 3 precomputed RDMs from the teacher network (24 comparisons over validation set of 5000 images). Due to the imperfect correlation between the *premature* and *mature* performances, doing a small post-search reranking step increases the chance of finding slightly better cell structures. We chose the top 10 discovered cells and trained them for 300 epochs on the training set and evaluated on the validation set (5k samples). Cell structure with the highest validation

Table 1: Comparison of premature performance and representational similarity measure in architecture search using RL and TPE algorithms. P: premature performance as validation score; P+TG: combined premature performance and RDMs as the validation score. Values are $\mu \pm \sigma$ across 3 search runs.

| Search Algorithm | RL | | | | TPE | |
|---|---|---|---|---|---|---|
| Search Space | 10 layer | | 20 layer | | 10 layer | 20 layer |
| # Epoch/Sample | 2 | 20 | 2 | 20 | 2 | 2 |
| Random - Best C100 Error (%) | 45.4± 2.5 | 41.3± 1.5 | 41.2± 1.8 | 38.3± 4.8 | 45.4± 2.5 | 41.2± 1.8 |
| P - Best C100 Error (%) | 41.0± 0.5 | 40.5± 0.4 | 37.5± 0.2 | 32.7± 0.9 | 42.5± 5.7 | 37.0± 3.0 |
| P+TG - Best C100 Error (%) | **38.3± 1.1** | **39.2± 0.9** | **33.2± 1.4** | **32.2± 0.8** | **37.6± 1.2** | **33.0± 2.4** |
| Performance Improvement (%) | 2.7 | 1.3 | 4.3 | 0.5 | 4.9 | 4 |

performance was then fully trained on the complete training set (50k samples) for 600 epochs using the procedure described in (Zoph et al., 2017) and evaluated on the test set[1].

Table 2: Performance of discovered cells on CIFAR10 and CIFAR100 datasets. *indicates error rates from locally training the network using the same training pipeline on 2-GPUs. [†]we did not further explore these hyperparameters because of compute limitations and adopted the values reported in (Zoph et al., 2017).

| Network | B | N | F | # Params | C10 Error | C100 Error | $M_1$ | $E_1$ | $M_2$ | $E_2$ | Cost |
|---|---|---|---|---|---|---|---|---|---|---|---|
| AmoebaNet-A | 5 | 6 | 36 | 3.2M | **3.34** | - | 20000 | 1.13M | 100 | 27M | 25.2B |
| NASNet-A | 5 | 6 | 32 | 3.3M | 3.41 (3.72*) | 17.88* | 20000 | 0.9M | 250 | 13.5M | 21.4-29.3B |
| PNASNet-5 | 5 | 3 | 48 | 3.2M | 3.41 (4.06*) | 19.26* | 1160 | 0.9M | 0 | 0 | 1.0B |
| ENAS | 5 | 6 | - | 4.6M | 3.54 | - | **310** | **50k** | **0** | **0** | **15.5M** |
| SAGENet | 5 | $6^†$ | $32^†$ | 6.0M | 3.66 | **17.42** | 1000 | 90K | 10 | 13.5M | 225M |
| SAGENet-sep | | | | **2.7M** | 3.88 | 17.51 | | | | | |

We compared our best found cell structure with those found using NAS (Zoph et al., 2017) and PNAS (Liu et al., 2017) methods on CIFAR-10, CIFAR-100, and Imagenet datasets (Tables 2 and 3). To rule out any differences in performance that might have originated from slight differences in training procedure, we used the same training pipeline to train our proposed network (SAGENet) as well as the as well as the two baselines (NASNet and PNASNet). We found that on all datasets, SAGENet performed on par with the other two baseline networks we had considered.

With regard to compactness, SAGENet had more parameters and FLOPS compared to NASNet and PNASNet due mostly to symmetric $7 \times 1$ and $1 \times 7$ convolutions. But we had not considered any costs associated with the number of parameters or the number of FLOPS when conducting the search experiments. For this reason, we also considered another version of SAGENet in which we replaced the symmetric convolutions with "$7 \times 7$ separable" convolutions (SAGENet-sep). SAGENet-sep had half the number of parameters and FLOPS compared to SAGENet and slightly higher error rates.

To compare the cost and efficiency of different search procedures we adopt the same measures as in (Liu et al., 2017). Total cost of search is computed as the total number of examples that were processed with SGD throughout the search procedure. This includes $M_1$ sampled cell structures that were trained with $E_1$ examples during the search and $M_2$ top cells trained on $E_2$ examples post-search to find the top performing cell structure. The total cost is then calculated as $M_1 E_1 + M_2 E_2$. While SAGENet performed on par to both NASNet and PNASNet top networks on all C10, C100, and Imagenet, the cost of search was about 100 and 4.5 times less than NASNet and PNASNet respectively (Table 2). A unique features of this cell is the large number of skip connections (both within blocks and across cells) (see Figure 5 in supp. material). Interestingly, at *mature* state our top architecture performed better than the teacher network (ResNet) on C10 and C100 datasets (96.34% and 82.58% on C10 and C100 for TG-SAGE as compared to 94.75% and 75.89% for our teacher-ResNet).

Table 3: Performance of discovered cells on Imagenet dataset in mobile settings.*indicates error rates from training all networks using the same training pipeline on 2-GPUs.

| Network | B | N | F | Top-1 Err* | Top-5 Err* | # Params (M) | FLOPS (B) |
|---|---|---|---|---|---|---|---|
| NASNet-A | 5 | 4 | 44 | 31.07 | 11.41 | 5.3 | 1.16 |
| PNASNet-5 | 5 | 3 | 56 | **29.92** | **10.63** | 5.4 | 1.30 |
| SAGENet | 5 | 4 | 48 | 31.81 | 11.79 | 9.7 | 2.15 |
| SAGENet-sep | | | | 31.9 | 11.99 | **4.9** | **1.03** |
| SAGENet-neuro | 5 | 4 | 40 | 31.77 | 11.72 | 6.1 | 1.59 |

## 4.4 USING CORTICAL MEASUREMENTS AS THE TEACHER NETWORK

As discussed earlier, the teacher network could be any network that is high-performing and its internal activations are partially observable. One such network is the primate brain that is both high

---

[1]our best network was ranked $3^{rd}$ and with a score difference of 0.004 with the top network.

performing in object categorization task and is partially observable through electrophysiological recording tools. As a proof of concept, we conducted an additional experiment in which we used neural measurements from macaque brain to guide the architecture search. In this setting, we used neural measurements from 296 neural sites from two macaque monkeys in response to 5760 images. Neural responses were collected from three anatomical regions along the ventral visual pathway (V4, posterior-inferior temporal (p-IT), and anterior inferior temporal (a-IT) cortex) in each monkey – a series of cortical regions in the primate brain that facilitate object recognition. The presented images contained 3D rendered objects placed on uncorrelated natural backgrounds and were designed to include large variations in position, size, and pose of the objects (see supplementary material). To allow the candidate networks to be more comparable to the brain measurements, we conducted the experiment on Imagenet dataset and trained each candidate network for one epoch using images of size $64 \times 64$. We used the same setup as in section 4.3 but this time with three RDMs generated from our neural measurements in each area (i.e. V4, p-IT, a-IT). We held out 50,000 of the images from the original Imagenet training set as the validation set that was used to evaluate the premature performance for the candidate networks. To further speed up the search, we removed the first 2 reduction cells in the architecture during the search.

After running the architecture search for 1000 samples, we picked the top 10 networks and fully trained them on imagenet for 40 epochs and picked the network with highest validation accuracy. We then trained this network on Cifar10, Cifar100, and Imagenet datasets and evaluated its performance on the test set. The best discovered network found this way (SAGENet-neuro) reached comparable performance levels to those we found in section 4.3. The best discovered network reached a validation error of 18.28%, 3.87%, and 31.77% on Cifar10, Cifar100, and Imagenet datasets respectively.

Although the number of neural sites that were included in this experiment was on the order of only few hundred, the representational description that these measurements provided were still informative enough to produce a meaningful guidance for the architecture search. However, we also acknowledge that there are many unexplored questions left regarding the effectiveness of neural measurements as constraints to inform and guide the search for higher performing artificial neural networks.

## 5  DISCUSSION AND FUTURE DIRECTIONS

We here demonstrate that, when the internal neural representations of a powerful teacher neural network are partially observable (such as the brain's neural network), that knowledge can substantially accelerate the discovery of high performing machine networks. We propose a new method to accomplish that acceleration (TG-SAGE) and demonstrate its ability using a previous state-of-the-art network as the teacher. Essentially, TG-SAGE jointly maximizes a model's *premature* performance and its representational similarity to those of a partially observable teacher network. With the architecture space and search settings tested here, we report a computational efficiency gain of $\sim 10\times$ in discovering CNNs for visual categorization. This gain in efficiency of the search (with maintained performance) was achieved without any additional constraints on the search space as in more efficient search methods like ENAS (Pham et al., 2018) or DARTS (Liu et al., 2018). We empirically demonstrated this by performing searches in several CNN architectural spaces. In addition, as a proof of concept, we here showed how limited measurements from the brain (neural population patterns of responses to many images) could be formulated as teacher constraints to accelerate the search for higher performing networks. However, it remains to be seen if larger scale neural measurements – which are obtainable in the near future – could achieve even better acceleration.

An important aspect of teacher guided architecture search relates to the metrics used for evaluating similarity of representational spaces. Here we used representational dissimilarity matrix to achieve this goal. However, we acknowledge that RDM might not be the most accurate or fastest metric for this purpose. Exploring other representational analysis metrics like Singular Vector Canonical Correlation Analysis (SVCCA) are an important direction we would like to pursue in the future.

Another interesting future direction would be to conduct the architecture search by iteratively substituting the teacher network with the best network discovered so far. This approach would make the procedure independent of the choice of the teacher network and make it possible to perform efficient search when good teacher architectures have not been deployed yet.

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

## SUPPLEMENTARY MATERIAL

### HYPERPARAMETER SEARCH WITH REINFORCEMENT LEARNING (RL)

We follow the method proposed by (Zoph & Le, 2017) to learn the probability of hyperparameter choices ($\mathcal{X} = x_1, x_2, ..., x_n$) that maximize the unknown but observable reward function $f : \mathcal{X} \rightarrow \mathbb{R}$. A 2-layer long short-term memory (LSTM) is used as the controller that chooses each hyperparameter in the network at every unrolling step. The LSTM network, models the conditional probability distribution of optimal hyperparameter choices as a function of all previous choices $P(x_j | x_1, x_2, ..., x_{j-1}, \theta)$ in which $\theta$ is the set of all tunable parameters in the LSTM network. Since a differentiable loss function is not known for this problem, usual maximum likelihood methods could not be used in this setting. Instead parameters are optimized through reinforcement learning based approaches (e.g. REINFORCE (Williams, 1992)) by increasing the likelihood of each hyperparameter choice according to the reward (score) computed for each sampled network (or a batch of sampled networks). Relative to (Zoph & Le, 2017), we made two modifications. First, since the order of dependencies between different hyperparameters in each layer/block is arbitrary, we ran the LSTM controller for one step per layer (instead of once per hyper-parameter). This results in shorter choice sequences generated by the LSTM controller and therefore shorter sequence dependencies. Second, we chose a Boltzman policy method for action selection to allow the search to continue the exploration throughout the search experiment. Hyperparameter values were selected according to the probability distribution over all action choices. Compared to $\epsilon$-greedy method, following the softmax policy reduces the likelihood of sub-optimal actions throughout the training.

For each hyperparameter, choice probability is computed using a linear transformation (e.g. $W_{K_h}, W_{N_{filters}}$) from LSTM output at the last layer ($h_l^2$) followed by a softmax. To reduce the number of tunable parameters and more generalization across layers, we used shared parameters between layers.

$$\hat{P}_{l,x} = \text{softmax}(W_t^T h_l^2) \tag{2}$$
$$l \in \{1, 2, ..., N_l\}$$
$$t \in \{K_h, K_w, N_{filters}, \text{stride}, \text{normalization}, \text{activation}\}$$

Probability distribution over possible number of layers is formulated as a function of the first output value of the LSTM ($\hat{P}_{N_l} = \text{softmax}(W_{N_l}^T h_0^2)$). In addition to layers' hyperparameters we also search over layers' connections. Similar to the approach taken in (Zoph & Le, 2017) we formulated the probability of a connection between layer $i$ and $j$ as a function of the state of the LSTM at each of these layers ($h_i^2, h_j^2$).

$$\hat{P}_{i,j}^c = \text{sigmoid}(W_{src}^T h_i^2 + W_{dst}^T h_j^2) \tag{3}$$

where $\hat{P}_{i,j}^c$ represents the probability of a connection between layer $i$ output to $j$'s input. $W_{src}$ and $W_{dst}$ are tunable parameters that link the hidden state of LSTM to probability of a connection existing between the two layers.

### HYPERPARAMETER SEARCH WITH TREE OF PARZEN ESTIMATORS (TPE)

Sequential Model-Based Optimization (SMBO) (Hutter et al., 2011) approaches are numerical methods used to optimize a given function $f : \mathcal{X} \rightarrow \mathbb{R}$. They are usually applied in settings where evaluating the function at each point is costly and it's important to minimize the number of evaluations to reach the optimal value. Various SMBO approaches were previously proposed (Bergstra et al., 2012b; Bardenet & Kegl, 2010) and some have been used for hyperparameter optimization in neural networks (Bergstra et al., 2011; 2012a; Liu et al., 2017). Bayesian SMBO approaches model the posterior or conditional probability distribution of values (scores) and use a criteria to iteratively suggest new samples while the probability distribution is updated to incorporate the history of previous sample tuples $(x, y)$ where $x = (x^{(1)}, ..., x^{(n)})$ is a sample hyperparameter vector and $y$ is the received score (or loss). Here we adopted Tree of Parzen Estimators (TPE) because of its intuitiveness and successful application in various domains with high dimensional spaces. Unlike most other Bayesian SMBO methods that directly model the posterior distribution of values $P(y|x)$, TPE

models the conditional distribution $P(x|y)$ with two non-parametric densities.

$$P(x|y) = \begin{cases} l(x) & \forall\, y \leq y^* \\ g(x) & \forall\, y > y^* \end{cases} \tag{4}$$

We consider $y$ to be the loss value which we are trying to minimize (e.g. error rate of a network on a given task). For simplicity, value of $y^*$ could be taken as some quantile of values observed so far ($\gamma$). At every iteration, TPE fits a kernel density estimator with Gaussian kernels to subset of observed samples with lowest loss value ($l(x)$) and another to those with highest loss ($g(x)$). Ideally we want to find $x$ that minimizes $y$. Expected Improvement (EI) is the expected reduction in $f(x)$ compared to threshold $y^*$ under current model of $f$. Maximizing EI, encourages the model to further explore parts of the space which lead to lower loss values and can be used to suggest new hyperparameter samples.

$$EI(x) = \int_{-\infty}^{y^*} (y^* - y)P(y|x)dy = \frac{\int_{-\infty}^{y^*} (y^* - y)P(x|y)P(y)dy}{P(x)} \tag{5}$$

Given that $P(y < y^*) = \gamma$ and $P(x|y) = l(x)$ for $y < y^*$, it has been shown (Bergstra et al., 2011) that EI would be proportional to $\left(\gamma + \frac{g(x)}{l(x)}(1-\gamma)\right)^{-1}$. Therefore the EI criterion can be maximized by taking samples with minimum probability under $g(x)$ and maximum probability under $l(x)$. For simplicity, at every iteration $n_d$ samples are drawn from $l(x)$ and the hyperparameter choice with lowest $g(x)/l(x)$ ratio is suggested as the next sample.

## EFFECT OF WEIGHTING TEACHER GUIDANCE ON PREDICTING MATURE PERFORMANCE

For each candidate model, we computed an overall "Teacher Guidance (TG)" score by averaging the best RDM scores for all teacher layers. The combined "P+TG" score was formulated as weighted sum of premature performance and TG score in the form of $P + \alpha TG$. We varied the $\alpha$ parameter and measured the change in correlation between the P+TG score and the mature performance (Figure 4). We observed that higher $\alpha$ led to larger gains in predicting the mature performance when models were trained only for few epochs ($\leq 2.5$ epochs). However, with more training, larger $\alpha$ values reduced the predictability.

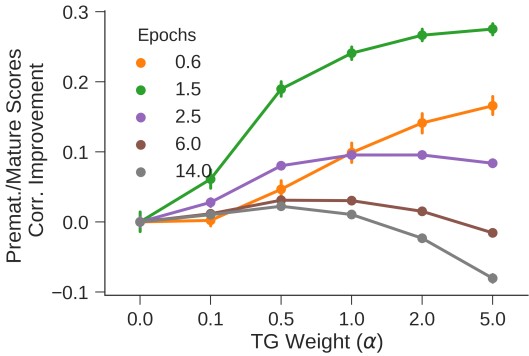

Figure 4: Effect of TG weight $\alpha$ on predicting the mature performance.

## ALTERNATIVE TEACHER NETWORK - NASNET

We examined the effect of choosing an alternative teacher network, namely NASNet and performed a set of analyses similar to those done on ResNet. We observed that similar to ResNet, early layers are better predictors of the mature performance during early stages of the training. With additional training, the premature performance becomes a better single-predictor of the mature performance but during most of the training the combined P+TG score best predicts the mature performance (Figure 5-left). We also varied the "TG" weight factor and found that compared to ResNet, higher $\alpha$ values led to increased gains in predicting the mature performance. $\alpha = 5$ was used to compute the P+TG scores shown in Figure 5.

Overall, we found that NASNet representations were significantly better predictors of mature performance for all evaluated time points during training when compared to ResNet (Figure 5-right).

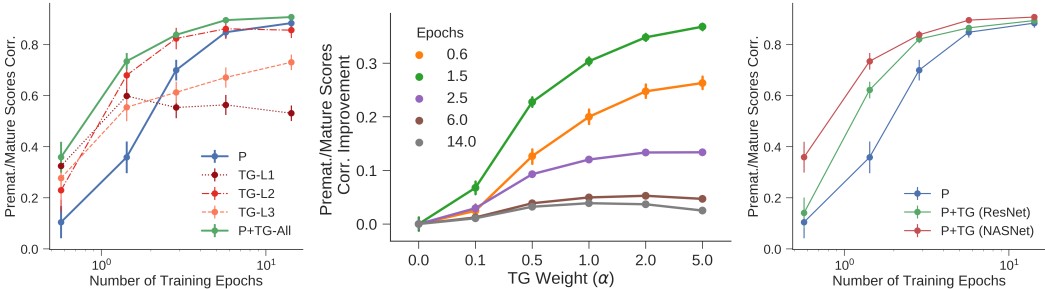

Figure 5: (left) Comparison of single layer and combined RDMs with premature performance as predictors of mature performance on NASNet. P+TG was computed using $\alpha = 5$. (middle) Gain in predicting the mature performance with varying TG weight. (right) Comparison of combined RDM scores using two alternative teacher models at various stages of training. $\alpha$ values of 1 and 5 were used for ResNet and NASNet respectively.

DATASETS AND PREPROCESSING

**CIFAR:** We followed the standard image preprocessing for CIFAR labeled dataset, a 100-way object classification task (He et al., 2015). Images were zero-padded to size $40 \times 40$. A random crop of size $32 \times 32$ was selected, randomly flipped along the horizontal axis, and standardized over all pixel values in each image to have zero mean and standard deviation of 1. We split the training set into training set (45,000 images) and a validation set (5,000 images) by random selection.

**Imagenet:** We used standard VGG preprocessing (Simonyan & Zisserman, 2014) on images from Imagenet training set. During training, images were resized to have their smaller side match a random number between 256 and 512 while preserving the aspect ratio. A random crop of size 224 was then cut out from the image and randomly flipped along the central vertical axis. The central crop of size 224 was used for evaluation.

DETAILS OF SEARCH ALGORITHMS

**RL Search Algorithm:** We used a two-layer LSTM with 32 hidden units in each layer as the controller. Parameters were trained using Adam optimizer (Kingma & Ba, 2014) with a batch size of 5. For all searches, the learning rate was 0.001, and the Adam first momentum coefficient was zero $\beta_1 = 0$. Gradients were clipped according to global gradient norm with a clipping value of 1 (Pascanu et al., 2012).

**TPE Search Algorithm:** We used the python implementation of TPE hyperparameter search from HyperOpt package (Bergstra et al., 2013). We employed the linear sample forgetting as suggested in (Bergstra et al., 2012a) and set the threshold $y^* = \sqrt{N}/4$ for the set of $N$ observed samples. Each search run started with 20 random samples and continued with TPE suggestion algorithm. At every iteration, $n_d = 24$ draws were taken from $l(x)$ and choice of hyperparameter $argmin_i g(x_i)/l(x_i)$ was used as the next sample (see section 3.3 in the main text).

EXPERIMENTAL DETAILS FOR SEARCH IN SPACE OF CONVOLUTIONAL NETWORKS

**Search Space:** Similar to (Zoph & Le, 2017) we defined the hyperparameter space as the following independent choices for each layer: $N_{filters} \in [32, 64, 128]$, $(K_{width}, K_{height}) \in [1, 3, 5, 7]$, $K_{stride} \in [1, 2]$, activation $\in [Identity, ReLU]$, normalization $\in [none, BN]$. In addition we searched over number of layers ($N_{layers} \in [1, N_L]$) and possible connections between the layers. In this space of CNNs, the input to every layer could have originated from the input image or the output of any of the previous layers. We considered two particular spaces in our experiments that differed in the value of $N_L$ (=10 or 20).

**CIFAR Training:** Selected networks were trained on CIFAR training set (45k samples) from random initial weights using SGD with Nesterov momentum of 0.9 for 300 epochs on the training set. The initial learning rate was 0.1 and was divided by 10 after every 100 epochs. Mature performance was then evaluated on the validation set (above).

EXPERIMENTAL DETAILS FOR SEARCH IN SPACE OF CONVOLUTIONAL CELLS

**Search Space:** We used the same search space and network generation procedure as in (Zoph et al., 2017; Liu et al., 2017) with the exception that we added two extra hyperparameters which could force each of the cell inputs (from previous cell or the one prior to that) to be directly concatenated in the output of the cell even if they were already connected to some of the blocks in the cell. This extra hyperparameter choice was motivated by the open-source implementation of NASNet at the time of conducting the search experiments that contained similar connections[2].

Each cell receives two inputs which are the outputs of the previous two cells. In early layers, the missing inputs are substituted by the input image. Each cell consists of $B$ blocks with a prespecified structure. Each block receives two inputs, an operation is applied on each input independently and the results are added together to form the output of the block. The search algorithm picks each of the operations and inputs for every block in the cell. Operations are selected from a pool of 8 possible choices: {identity, $3 \times 3$ average pooling, $3 \times 3$ max pooling, $3 \times 3$ dilated convolution, $1 \times 7$ followed by $7 \times 1$ convolution, $3 \times 3$ depthwise-separable convolution, $5 \times 5$ depthwise-separable convolution, $7 \times 7$ depthwise-separable convolution}.

**Imagenet Training:** For our Imagenet training experiments, we used a batch size of 128 images of size $224 \times 224$ pixels. Each batch was divided between two GPUs and the gradients computed on each half were averaged before updating the weights. We used an initial learning rate of 0.1 with a decay of 0.1 after every 15 epochs. Each network was trained for 40 epochs on the Imagenet training set and validated on the central crop for all images from Imagenet validation. No dropout or drop-path was used when training the networks. RMSProp optimizer with a decay rate of 0.9 and momentum rate of 0.9 was used during training and gradients were normalized by their global norm when the norm value exceeded a threshold of 10. L2-norm regularizer was applied on all trainable weights with a weight decay rate of $4 \times 10^{-5}$.

**CIFAR Training:** The networks were trained on CIFAR10/CIFAR100 training set including all 50,000 samples for 600 epochs with an initial learning rate of 0.025 and a single period cosine decay (Zoph et al., 2017). We used SGD with Nesterov momentum rate of 0.9. We used L2 weight decay on all trainable weights with a rate of $5 \times 10^{-4}$. Gradient clipping similar to that used for Imagenet and a threshold of 5 was used.

**Best Discovered Convolutional Cell:** Figure 6 shows the structure of the best discovered cell. Only four (out of ten) operations contain trainable weights and there are several bypass connections in the cell.

NEURAL MEASUREMENTS FROM MACAQUE MONKEYS

We used a dataset of neural spiking activity for a population of 296 neural sites in two awake behaving macaque monkeys in response to 5760 images (Yamins et al., 2014). Neural data were collected using parallel microelectrode arrays that were implanted chronically on the cortical surface in area V4 and IT. Fixating animals were presented with images for 100ms, and the neural response patterns were obtained by averaging the spike counts in the time window of 70-170ms post stimulus onset. To enhance the signal-to-noise ratio, each image was presented to each monkey between 21-50 times and the average response pattern across all presentation were considered for each image. The 296 recorded sites were partitioned into three cortical regions (V4, posterior-IT, and anterior-IT) and a RDM was calculated for each region.

The image set consisted of a total of 5760 images. Each image contained a 3D rendered object placed on an uncorrelated natural background. The rendered objects were selected from a battery

---

[2]available at `https://github.com/tensorflow/models/blob/376dc8dd0999e6333514bcb8a6beef2b5b1bb8da/research/slim/nets/nasnet/nasnet_utils.py`

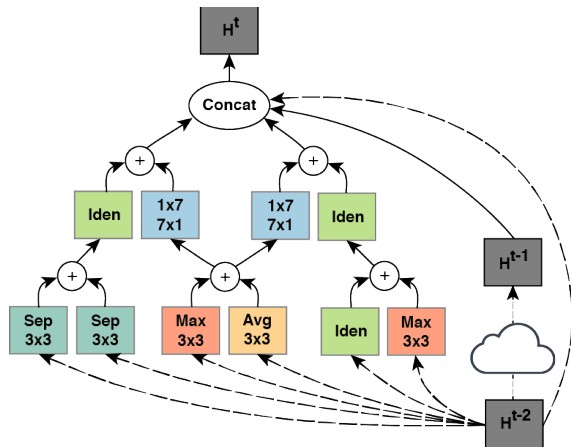

Figure 6: SAGENet - Structure of the best cell discovered during the search with TG-SAGE.

of 64 objects from 8 categories (animals, boats, cars, chairs, faces, fruits, planes, and tables) with 8 objects per category. The images were generated to include large variations in position, size, and pose of the objects and were shown within the central $8°$ of monkeys' visual field. Some example images are illustrated in Figure-7.

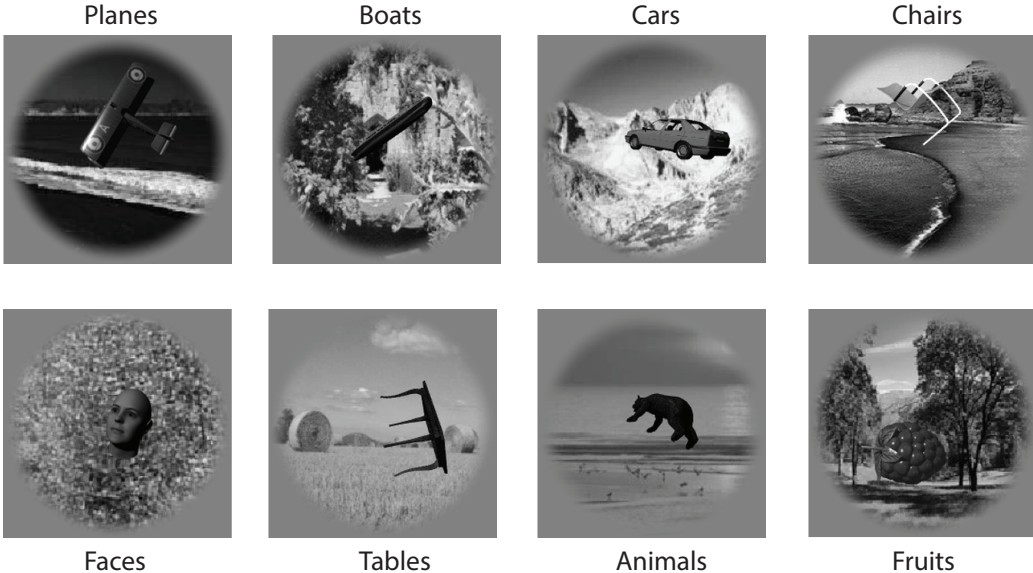

Figure 7: Example images from each of the eight object categories that were used to record neural responses.

IMPLEMENTATION DETAILS

Because of heavy computational load associated with training neural networks and in particular in large-scale model training, we needed a scalable and efficient framework to facilitate the search procedure. We implemented our proposed framework in four main modules: (i) explorer, (ii) trainer, (iii) evaluator, and (iv) tracker. The explorer module contained the search algorithm. The trainer module optimized the parameters of the proposed architecture on an object recognition task using a large-scale image dataset. Once the training job was complete the evaluator module extracted the network activations in response to a set of predetermined image-set and assessed the similarity of representations to the bank of neural and behavioral benchmarks (derived from human and non-

human primates). The tracker module consisted of a database which tracked the details and status of every proposed architectures and acted as a bridge between all three modules. During the search experiments, the explorer module proposes new candidate architectures and records the details in the database (tracker module). It also continuously monitors the database for newly evaluated networks. Upon receiving adequate number of samples (i.e. when a new batch is complete), it updates its parameters. Active workers periodically monitor the database for newly added untrained models, and train the architecture on the prespecified dataset. After the training phase is completed, the evaluator module extracts the features from all layers in response to the validation set and computes the premature-performance and RDM consistencies and writes back the results in the database. The trainer and evaluator modules are then freed up to process new candidate networks. This framework enabled us to run many worker programs on several clusters speeding up the search procedure. An overview of the implemented framework is illustrated in Figure 8. Experiments reported in this paper were run on three server clusters with up to 40 GPUs in total.

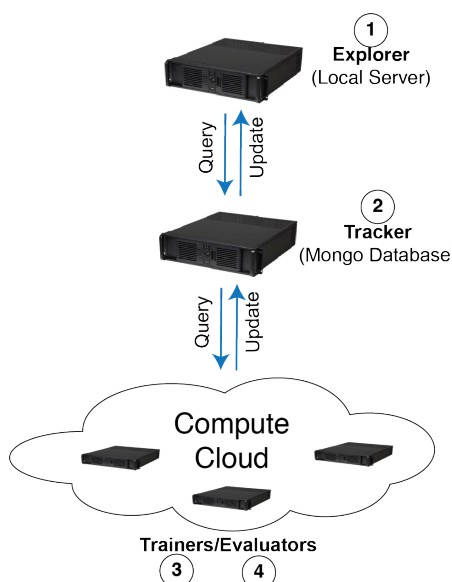

Figure 8: Implementation of a distributed framework for conducting architecture search.

