# OpenReview forum: "Teacher Guided Architecture Search"
_ICLR.cc/2019/Conference_

### Official Review · AnonReviewer1 · 2018-11-02
**confusingly written paper that also lacks some intuition**

**Rating:** 6
**Confidence:** 4

**Review:**

The paper proposes a new performance metric for neural architecture search based on the similarity of internal feature representations to a predefined fixed teacher network.


The idea to not only use performance after each epochs as a signal to guide the search procedures is sensible.
However, the description of the  proposed method is somewhat confusing and lacks some intuition:

1) Why should a new architecture necessarily mimic the internal representation of the teacher network? Wouldn't the best configuration simply be an exact copy of the teach network?
  A well-performing networks could still have a low TG score, simply because its internal representation does not match the teacher layer-wise.

2) Probably, in the most scenarios on new tasks, a teacher network is not available. This somewhat contradicts the intention of NAS / AutoML, which aims to automatically find well-performing networks without any human intervention or prior knowledge.

3) It is unclear to me how to compute the correlation of RDMs in cases where the architecture space is not as structured as in the paper (=either just sequential models or cell search space)

4) Figure 1, middle: while the overall correlation of 0.62 is ok, it seems that the correlation for high-performing models (=the region of interest),say P+TG > 0.5, is rather small/non-existing



Minor comments:

 - Section 3.3 first sentence says: "Sequential Model-Based Optimization approaches are greedy numerical method" : this is not correct since they use an acquisition function to pick new candidates which trades-off exploration and exploitation automatically. Furthermore, under some assumptions, Bayesian optimization converges to the global optimum.

- I think, arguing that one can use the human brain as a teacher network is a bit of a stretch. Also, the authors do not give any explanation how that could be realized.

- Section 3.3 says that TPE fits a Gaussian Mixture Model, however, it actually fits a kernel density estimator.

---

> ### Author Response · Authors · 2018-11-25
> **Clarified the method and added a section regarding the plausibility of brain-guided architecture search**
>
> We thank the reviewer for detailed feedback and questions. We provide specific responses to each point below.
>
> 1. "Why should a new architecture necessarily mimic ...":
> While the exact feature space that each specific network converges to is unique to that network, there are many similarities between the feature spaces in different networks. For example, it has previously been shown that object-level representations (like those obtained from object-level RDM that is also used in our work) at higher layers of several convolutional neural networks are largely similar to each other and also to measurements from the brain [1]. Our approach encourages the representational similarity to that of the teacher model. However, it should be noted that the similarity comparison is done between the candidate network at an early stage during the training and the fully trained teacher network. Therefore, we would argue that it would be unlikely that the representational similarity score would lead to finding an exact copy of the teacher network during the search. Instead, it encourages to find networks that are able to construct representations early during their training that are most similar to those of a teacher network found after orders of magnitude more training.
>
> 2. "Probably, in the most scenarios on new tasks, a teacher network is not available...": We agree with the reviewer that the goal of NAS/AutoML is to discover good networks with minimal dependence on the human intervention/knowledge. Our approach requires measurements from a high-performing network to function. As was mentioned in response to the previous comment, current state-of-the-art models bear remarkable similarity to those from brain measurement [1]. In addition, it has further been shown that performance on object-recognition tasks is highly correlated with the representation similarity between the network and the brain [2]. These two observations constitute the core of our approach. We consider the brain as a high-performing (biological) neural network with unknown architectural parameters and partially observable activity. We discussed the plausibility of this approach in more detail in response to the next comment.
>
> 3. "I think, arguing that one can use the human brain as a teacher...":
> As of now, in many labs including our own, simultaneous neural measurements on the order of several hundred neurons are possible and the emerging new technology is making larger population recordings possible (please see [3, 4] for references). As a proof of concept, we also conducted another search experiment guided by measurements from macaque monkey brains. We discussed the outcomes of this search in the newly added section 4.4. Briefly, we used neural population (n=296) response patterns to 5760 images to produce object-level RDMs. We then conducted a search experiment on the space of convolutional cells with the goal of maximizing the combined P+TG score. The best discovered model achieved similar performance levels to those found with a ResNet teacher model.
>
> 4. "It is unclear to me how to compute ...": RDM can be computed for any representation space in response to a set of inputs. Given a feature matrix, we first compute the average response vectors for each object category by computing the average response over all inputs belonging to the same object category. Then we compute the Pearson correlation between average response vectors of each pair of objects to form the RDM. We added a new figure (Figure-1) and added a more detailed explanation in section 3.1.
>
> 5. "Figure 1, middle: while the ...": The left and middle panels in Figure 1 were plotted for training after 1.5 epochs. We updated this figure with the scatter plot for 2 epochs of training that was used during our experiments. It is clear from these plots that by around 2 epochs of training the correlation between "P+TG" and Mature performance is both higher and more uniform across the range of scores.
>
> 6. "Section 3.3 first sentence says: "Sequential Model-Based Optimization approaches are greedy numerical method...": We agree with the reviewer that because of the sampling function these methods are not necessarily greedy. We dropped the term "greedy" from the statement.
>
> 7. "Section 3.3 says that TPE fits ...": We thank the reviewer for bringing this issue up. We changed this to "TPE fits a kernel density estimator with Gaussian kernels"
>
> [1] Cadieu, et al. (2014). Deep Neural Networks Rival the Representation of Primate IT Cortex for Core Visual Object Recognition. PLoS Computational Biology, 10(12).
> [2] Yamins, et al. (2014). Performance-optimized hierarchical models predict neural responses in higher visual cortex. Proceedings of the National Academy of Sciences, 111(23), 8619–8624.
> [3] Stevenson, Ian H., and Konrad P. Kording. "How advances in neural recording affect data analysis." Nature neuroscience 14.2 (2011): 139.
> [4] https://stevenson.lab.uconn.edu/scaling/

---

> ### Comment · AnonReviewer1 · 2018-11-29
> **After rebuttal comment**
>
> I thank the authors for addressing my comments and concerns. The rebuttal has certainly clarified the paper and I will increase my score accordingly. However, the experimental section needs further clarifications, hence only the weak accept.
>
>
> - As it is well known from the literature[1, 2, 3], just using the premature performance as a surrogate for the mature performance leads to poor prediction, particularly in the early stages of the training. Hence, recent methods try to predict the learning curve of the architectures [1, 2, 3]. The approach would be more convincing, if the paper could show a comparison to some of these methods.
>
> - It seems that authors only use CIFAR-100 because of its larger number of classes that provide a better RDM. Would the proposed score also work well for datasets with smaller number of classes such as CIFAR-10?
>
> - The authors use only one teacher network for all the experiments. It would be interesting to know how much the choice of the teacher networks affects the final performance.
>
>
> [1] Tobias Domhan, Jost Tobias Springenberg, Frank Hutter: Speeding Up Automatic Hyperparameter Optimization of Deep Neural Networks by Extrapolation of Learning Curves. IJCAI 2015
> [2] Bowen Baker, Otkrist Gupta, Ramesh Raskar, Nikhil Naik: Practical Neural Network Performance Prediction for Early Stopping. CoRR abs/1705.10823 (2017)
> [3] Aaron Klein, Stefan Falkner Jost Tobias Springenberg, Frank Hutter: Learning Curve Prediction with Bayesian Neural Networks, ICLR 2017

---

> > ### Author Response · Authors · 2018-12-02
> > **RDM resolution affects the gain in efficiency**
> >
> > We thank the reviewer for providing a second round of feedback on our work. We responded to each comment below:
> >
> > 1. "As it is well known from the literature[1, 2, 3], just using the premature performance as a surrogate for the mature performance leads to poor prediction,...":
> > We agree with the reviewer that comparing the efficiency of our method with some of these alternative methods is valuable. Experiments reported in the current version of our paper were selected to enable us to compare our method against a different battery of search methods [1-3] and unfortunately are not suitable to present a direct comparison with the papers suggested by the reviewer. Given the time limit for the review period it is unlikely that we can finish these experiments in time. Nonetheless, we like to note that all three of the studies cited by the reviewer require generating large datasets (on the order of several hundred models) to construct a predictive model of the mature performance. Hence, we anticipate that our method would still be more efficient than the competing methods because of no computational overhead for creating such model, but we also acknowledge that only a direct comparison between the methods could enable us to make that claim.
> >
> > 2. "It seems that authors only use CIFAR-100 because of its larger number of classes that provide a better RDM... ":
> > Regarding applicability on CIFAR-10 dataset, we performed a correlation analysis on CIFAR-10 dataset similar to that reported in Figure-2 fro CIFAR-100. We found that the correlation between object-level RDMs and mature-performance on C10 dataset was much lower than those we found for C100 and P+TG score was only marginally better than performance only measure during early stages of training. Since the main difference between these two datasets is in the number of classes, we conjectured that this observation is potentially because of lower resolution RDMs on C10 dataset. However, another possibility that we have only partly explored so far is to use image-level RDMs that are computed over specific images instead of objects. Image-level RDMs produce the same resolution regardless of the number of classes.
> >
> > 3. "The authors use only one teacher network for all the experiments. ...":
> > We had conducted an analysis reported in supplementary material (ALTERNATIVE TEACHER NETWORK - NASNET) in which we performed the correlation analysis on a second teacher network (NASNet) with higher mature performance. We found that although the P+TG correlation with mature performance was higher for this teacher network, the enhancement in correlation over ResNet teacher network was negligible when the candidate networks were trained for about 2 epochs or more (Figure-5).
> >
> >
> > [1] Zoph, B., Vasudevan, V., Shlens, J., & Le, Q. V. (2017). Learning Transferable Architectures for Scalable Image Recognition, 10. Retrieved from http://arxiv.org/abs/1707.07012
> >
> > [2] Liu, C., Zoph, B., Shlens, J., Hua, W., Li, L.-J., Fei-Fei, L., … Murphy, K. (2017). Progressive Neural Architecture Search. Retrieved from http://arxiv.org/abs/1712.00559
> >
> > [3] Pham, H., Guan, M. Y., Zoph, B., Le, Q. V., & Dean, J. (2018). Efficient Neural Architecture Search via Parameters Sharing. Retrieved from https://arxiv.org/abs/1802.03268

---

### Official Review · AnonReviewer3 · 2018-11-02
**Exciting idea but a comparison to methods with similar intent is missing**

**Rating:** 5
**Confidence:** 4

**Review:**

This work tries to accelerate neural architecture search by reducing the computational workload spend to evaluate a single architecture. Given a network with premature performance (i.e. one trained only for few epochs), it tries to compute the mature performance (accuracy achieved when training the network to completion). A score for each architecture is given by "P+TG" which considers the validation accuracy and the similarity of instance representations at different layers to a given teacher network. In an experiment against using the validation accuracy as a score only, they achieve higher validation accuracies on CIFAR-10/100. In a comparison against NASNet and PNASNet, their experiments indicate higher validation accuracy in orders of magnitudes faster run time.

This is a well-written paper with an innovative idea to forecast the mature performance. I am not aware of any other work using the internal representation in order to obtain that. However, there is plenty of other work aiming at accelerating the architecture search by early stopping based on premature performance. Hyperband [1] uses the premature performance (considered in this paper), many others try to forecast the learning curve based on the partial observed learning curve [2,3]. ENAS [4] learns shared parameters for different architectures. Predictions on batches are then used as a surrogate. SMASH [5] uses a hypernetwork to estimate the surrogate. While this work was partly mentioned, neither the close connection to this work is discussed nor serves any of these methods as a baseline.
Typically, the premise of automatic neural architecture search is that the user does not know deep learning well. However, the authors assume that a teacher exist which is an already high performing architecture. It is unclear whether this is a realistic scenario in real-life. How do you find the teacher's architecture in the first place? Does the method also work in cases where the teacher is performing poorly?
The P+TG score depends on a hyperparameter alpha. There is no empirical evidence supporting that alpha can be fixed once and applied for any arbitrary dataset. So I have concerns whether this approach is able to generalize over different datasets. The experiment in the appendix confirms this by showing that a badly chosen alpha will lead to worse performance.
Additionally, I think the paper would benefit from spending more space on explaining the proposed method. I would rather see more details about RDM and maybe a nice plot explaining the motivation rather than repeating NAS and TPE.

[1] Lisha Li, Kevin G. Jamieson, Giulia DeSalvo, Afshin Rostamizadeh, Ameet Talwalkar: Hyperband: A Novel Bandit-Based Approach to Hyperparameter Optimization. Journal of Machine Learning Research 18: 185:1-185:52 (2017)
[2] Tobias Domhan, Jost Tobias Springenberg, Frank Hutter: Speeding Up Automatic Hyperparameter Optimization of Deep Neural Networks by Extrapolation of Learning Curves. IJCAI 2015: 3460-3468
[3] Bowen Baker, Otkrist Gupta, Ramesh Raskar, Nikhil Naik: Practical Neural Network Performance Prediction for Early Stopping. CoRR abs/1705.10823 (2017)
[4] Hieu Pham, Melody Y. Guan, Barret Zoph, Quoc V. Le, Jeff Dean: Efficient Neural Architecture Search via Parameter Sharing. ICML 2018: 4092-4101
[5] Andrew Brock, Theodore Lim, James M. Ritchie, Nick Weston: SMASH: One-Shot Model Architecture Search through HyperNetworks. CoRR abs/1708.05344 (2017)

---

> ### Author Response · Authors · 2018-11-25
> **Updated the review of past work, clarified the methods and added an experiment regarding the plausibility of brain-guided architecture search**
>
> We thank the reviewer for the detailed review and provide responses to each comment below.
>
> 1. "...there is plenty of other work aiming at accelerating the architecture search by early stopping based on premature performance.": We acknowledge that several of these important previous works were missing in our manuscript. To address this, we added a paragraph in section 2 to discuss these approaches and compare them to ours. We also updated Table-2 with more baseline methods including ENAS that were previously applied to a similar cell-based architecture space. In Table-2, we only included methods that were previously tested on the same search space of convolutional networks.
>
> 2. "How do you find the teacher's architecture in the first place?": By definition, our method requires a teacher to operate. As stated in the abstract and introduction sections, our motivation for this work was to ultimately use measurements from primate brain as the "teacher" in this setup. As of now, in many labs including our own, simultaneous neural measurements on the order of several hundred neurons are possible and the emerging new technology is making larger population recordings possible (please see [1, 2] for references). Moreover, as a proof of concept, we conducted another search experiment guided by measurements from macaque monkey brains. We discussed the outcomes of this search in the newly added section 4.4.
>
> 3. "Does the method also work in cases where the teacher is performing poorly?": We examined mature/premature correlations for two alternative teacher networks (i.e. ResNet and NASNet). These networks had a relatively large difference in their mature performance (75 vs. 82). We observed that features from the higher performing teacher model (NASNet) were better predictors of mature performance (Figure 4-right panel). While here we tested only two teacher models, it is plausible to think that lower performing teacher networks would deliver weaker prediction of mature performance and therefore lesser gain in search efficiency.
>
> 4. "The P+TG score depends on a hyperparameter alpha. There is no empirical evidence supporting that alpha can be fixed once and applied for any arbitrary dataset. So I have concerns whether this approach is able to generalize over different datasets. The experiment in the appendix confirms this by showing that a badly chosen alpha will lead to worse performance.": Parameter alpha dictates the weight on the "TG" portion of the score compared to the premature-performance. It is true that this parameter affects the expected gain in correlation over the "performance-only" search. However, most alpha values produce an acceptable gain over the "performance-only" during the early stage of training. On the contrary, as in the later stages of training the premature-performance becomes a stronger predictor of the mature-performance, TG signal becomes irrelevant and only smaller values of alpha lead to marginal gain in correlation. In this work, our focus was on the early stages of training in which the gain is mostly robust to the choice of alpha (e.g. alpha = 1-5).
>
> 5. "Additionally, I think the paper would benefit from spending more space on explaining the proposed method.": We agree with the reviewer on the need for providing more details about the RDM method and how it is used to guide the architecture search. For this, we moved the RL and TPE search methods to the supplementary material, added a section on RDM in methods, provided more information about how RDM is used to guide the search and added a new figure (Figure-1) explaining RDM and motivating the method.
>
> [1] Stevenson, Ian H., and Konrad P. Kording. "How advances in neural recording affect data analysis." Nature neuroscience 14.2 (2011): 139.
>
> [2] https://stevenson.lab.uconn.edu/scaling/

---

> > ### Comment · AnonReviewer3 · 2018-11-27
> > **Why would RDM work for Neural Networks?**
> >
> > The authors significantly changed the original paper.
> >
> > As far as I understood, the RDM is computed on the intermediate representations of the neural network. However, given a CNN it is possible to rearrange the filters without the networks predictions. RDM however does not account for that because this is not the case for the human brain. So what is your explanation why RDM works?
> >
> > How exactly do you compute the RDM between a CNN and sensor readings of a monkey brain? I'm not a neuroscientist but this does not sound right to me at all. Why should we expect similarities between brain activation patterns and intermediate representations of a CNN?
> >
> > I'd really like to see an experiment which compares the proposed method to methods predicting the mature performance based on observed premature performance.

---

> > > ### Author Response · Authors · 2018-11-28
> > > **RDMs allow comparison between representational spaces regardless of their dimensionality**
> > >
> > > Thanks for the comment. There might be some misunderstanding regarding how RDM is computed and how it allows us to compare representational spaces in biological and artificial systems. We responded to each comment below:
> > >
> > > 1. "As far as I understood, the RDM is computed on the intermediate representations of the neural network. However, given a CNN it is possible to rearrange the filters without the networks predictions. RDM however does not account for that because this is not the case for the human brain. So what is your explanation why RDM works?":
> > > RDM characterizes a representational space with respect to the similarity of responses to inputs of different categories. As an example consider a simple case that we are only considering two categories of inputs (e.g. cats and fish). We show the neural network an image of a “cat” and an image of a “fish”. For a given layer, we compute the activations of “N” features in that layer in response to these two images. We rearrange these features into a vector (i.e. flatten the tensor into a vector) that constitutes the response vector to each image at the given layer. Note that we maintain the same order of features in the response vector for all images. In other words, feature “i” in the response vectors for “cat” and “fish” images contain activations from the same feature in the network.
> > > Having these response vectors, to construct the RDM, we find the correlation between the response vectors for each pair of images. In this case the possible pairs are "cat to cat", "fish to fish" and "cat to fish". The correlation for the trivial cases ("cat to cat", "fish to fish") are equal to one and for the nontrivial case ("cat to fish"), it quantifies how similar the response vectors are at the given layer.
> > > For the specific case of “rearranging the filters in CNN” (as the reviewer suggests), the RDM would not change as long as the same feature-order is maintained when collecting the response vectors for all the images. We hope this explanation has clarified the RDM procedure. A more thorough explanation is given in [1].
> > >
> > > 2. "How exactly do you compute the RDM between a CNN and sensor readings of a monkey brain? I'm not a neuroscientist but this does not sound right to me at all. Why should we expect similarities between brain activation patterns and intermediate representations of a CNN?":
> > > For the case of measuring the cortical activity, you can think of these measurements as subsampled features from the full set of features (neurons) in a specific brain region (similar to a layer in the network). Because these measurements are recorded using chronic implants, each neural site records from the same neuron in the brain in response to all the presented images. We use RDM to characterize the similarities between (neural) responses to images from different categories.
> > > To compare the similarity of the representational spaces in brain and CNN, we first compute the RDM for each space (one layer in CNN and one region in the brain), and then calculate the similarity between the RDMs by flattening both matrices and finding the correlation between the two vectors.
> > > The RDM similarity between neural responses in area V4 and IT and CNN features using RDM metric have been extensively studies previously[2, 3, 4]. Specially, Please see Fig. 4 in [2] and Fig. 7 in [3] that demonstrate and quantify the similarity of RDMs between the brain and several classes of CNNs.
> > >
> > > 3. "I'd really like to see an experiment which compares the proposed method to methods predicting the mature performance based on observed premature performance.":
> > > The experiments in sec 4.1 and results reported in Table-1 and Figure-3 were aimed at making a fair comparison between the two methods as the reviewer suggests (i.e. premature performance versus P+TG).
> > >
> > > [1] Kriegeskorte, N. (2008). Representational similarity analysis – connecting the branches of systems neuroscience. Frontiers in Systems Neuroscience, 2, 1–28. https://doi.org/10.3389/neuro.06.004.2008
> > >
> > > [2] Yamins, D. L. K., Hong, H., Cadieu, C. F., Solomon, E. A., Seibert, D., & DiCarlo, J. J. (2014). Performance-optimized hierarchical models predict neural responses in higher visual cortex. Proceedings of the National Academy of Sciences, 111(23), 8619–8624. https://doi.org/10.1073/pnas.1403112111
> > >
> > > [3] Cadieu, C. F., Hong, H., Yamins, D. L. K., Pinto, N., Ardila, D., Solomon, E. A., … DiCarlo, J. J. (2014). Deep Neural Networks Rival the Representation of Primate IT Cortex for Core Visual Object Recognition. PLoS Computational Biology, 10(12). https://doi.org/10.1371/journal.pcbi.1003963
> > >
> > > [4] Khaligh-Razavi, S. M., & Kriegeskorte, N. (2014). Deep Supervised, but Not Unsupervised, Models May Explain IT Cortical Representation. PLoS Computational Biology, 10(11). https://doi.org/10.1371/journal.pcbi.1003915

---

### Official Review · AnonReviewer2 · 2018-11-03
**Interesting idea and explorations**

**Rating:** 6
**Confidence:** 4

**Review:**

They propose a method of accelerating Neural Architecture Search by using a Teacher Network to predict (combined with immature performance)  the mature performance of candidate architectures, allowing them to train these candidates an order of magnitude faster (e.g. 2 epochs rather than 20).

It achieves 10x speed up in NAS. Interesting exploration of what is predictive of mature performance at different epochs. For example, TG(L1) is most predictive up to epoch 2, TG(L2/L3) are most predictive after, but immature performance + similarity to teacher (TG) is most predictive overall.

It has a very well-written related work section with a clear story to motivate the work.

The baselines in Table 2 should be updated. For example, NASNet-A on CIFAR10 (https://arxiv.org/pdf/1707.07012.pdf) reports an error of 3.41 without cutout and 2.65 with cutout, while the #Params is 3.3M. A more fair comparison should include those as baselines.

The experiments only consider 10 and 20 layer ConvNet.

The paper has lots of typos and missing articles or verbs.

---

> ### Author Response · Authors · 2018-11-25
> **Updated results tables and improved readability**
>
> We thank the reviewer for the constructive feedback and responded to specific comments below.
>
> 1. "The baselines in Table-2 should be updated.": We would like to note that the numbers reported in Table-2 were derived from training the open source implementation of NASNet and PNASNet using the same pipeline used to train our model (SAGENet). The reason for doing so was to discard the potential differences in performance values due to training procedures. However, we acknowledge that these values could be confusing for the reader and hence, we updated Table-2 and included both the original and retrained performances for these two networks in Table-2. In addition we included the performance values for several other related methods in tables 2 for the sake of completeness.
>
> 2. "The experiments only consider 10 and 20 layer ConvNet.": We used convolutional neural networks of depth 10 and 20 in our experiments in section 4.1 to motivate our approach. In addition to that, we did an additional experiment on a cell-based search space in the experiment in section 4.3 and the newly added section 4.4. The cell-based search space considered in section 4.3 and 4.4, contains diverse architectures with a wide range of variation with respect to depth. For instance, for a cell repetition value of N=5, a cell with 5 sequential sep-conv blocks could potentially produce a network of up to 210 layers deep.
>
> 3. "The paper has lots of typos and missing articles or verbs.": We thank the reviewer for bringing up this issue. We have gone over the manuscript and corrected the misspellings and missing articles/verbs.

---

### Author Response · Authors · 2018-11-25
**Summary of main changes in the revised submission**

We thank all the anonymous reviewers for their thoughtful comments. We made several changes to the manuscript to address the concerns that are listed below. We believe that these changes have significantly improved the quality of manuscript and made many of the details more clear compared to the original submission.

1. Added Figure-1 to motivate the approach and clarify the RDM metric.
2. Added a more thorough review of efficient architecture search methods in section 2.
3. Updated table 2 and 3 -- added more baseline methods.
4. Added section 4.4 -- architecture search guided by neural measurements from macaque monkey brain

---

### Meta-Review · Area_Chair1 · 2018-12-14
**motivation could be improved**

**Confidence:** 5
**Recommendation:** Reject

**Metareview:**

The authors propose to accelerate neural architecture search by using feature similarity with a given teacher network to measure how good a new candidate architecture is. The experiments show that the method accelerates architecture search, and has competitive performance. However, both Reviewers 1 and 3 noted questionable motivation behind the approach, as the method assumes that there already exists a strong teacher network in the domain where we architecture search is performed, which is not always the case. The rebuttal and the revised version of the paper addressed some of the reviewers' concerns, but overall the paper remained below the acceptance bar. I suggest that the authors further expand the evaluation and motivate their approach better before re-submitting to another venue.